# Knowledge, Attitudes, and Practices of Healthcare Providers Towards Advance Directive for COPD Patients in Riyadh, Saudi Arabia

**DOI:** 10.3390/healthcare13070771

**Published:** 2025-03-30

**Authors:** Rayan A. Qutob, Abdullah Alaryni, Yousef Alammari, Mohanad Khalid Almaimani, Abdullah Alghamdi, Abdulwahed Abdulaziz Alotay, Mohammad A. Alhajery, Fahad Ali Faqihi, Yassir Daghistani, Khalid I. AlHussaini, Saud Aldeghaither, Amal Alamri, Buthaina Alsharif, Hassan Alshamrani, Elaf Mubarak

**Affiliations:** 1Department of Internal Medicine, College of Medicine, Imam Mohammad Ibn Saud Islamic University (IMSIU), Riyadh 11623, Saudi Arabia; al3raini@hotmail.com (A.A.); yalammari@gmail.com (Y.A.); dr.alhomrani@gmail.com (A.A.); aaalotay@imamu.edu.sa (A.A.A.); maalhajery@imamu.edu.sa (M.A.A.); kialhussaini@imamu.edu.sa (K.I.A.); 2Department of Medicine, University of Jeddah, Jeddah 23218, Saudi Arabia; malmaimani@uj.edu.sa (M.K.A.); ydaghistani@uj.edu.sa (Y.D.); 3Department of Critical Care Medicine, Dr. Sulaiman Al Habib Medical Group Holding Company, Riyadh 11372, Saudi Arabia; fahad.faqihi@drsulaimanalhabib.com; 4Department of Critical Care Medicine, Prince Mohammed Bin Abdulaziz Hospital, Riyadh 14214, Saudi Arabia; aldeghaithers@pmah.med.sa; 5College of Medicine, Imam Mohammad Ibn Saud Islamic University (IMSIU), Riyadh 11623, Saudi Arabia; amal.n1alamri@gmail.com (A.A.); buthainaalsharif@gmail.com (B.A.); hassan.m.alshamrani@gmail.com (H.A.); elafnawafm@gmail.com (E.M.)

**Keywords:** advance directive, attitude, COPD, healthcare providers, knowledge, practice

## Abstract

**Background:** Chronic obstructive pulmonary disease (COPD) is a significant burden in Saudi Arabia. Improving the attitudes, awareness, and knowledge of healthcare providers toward advance directives and/or advanced care planning (ACP) can increase the use of advance directives. This study aims to investigate healthcare providers’ knowledge, attitudes, and practices concerning advance directives for COPD patients in Riyadh, Saudi Arabia. **Methods:** This cross-sectional study was employed to assess the knowledge, attitudes, and practices of healthcare providers towards ACP for COPD patients in Riyadh between June and December 2024. The questionnaire was adapted from previous research. Multiple logistic regression was performed to assess the factors associated with good knowledge and positive attitude. **Results:** A total of 268 participants were included in the analysis. The total mean of knowledge score was 6.96 ± 2.22 out of 12. A total of 161 participants (60.1%) had a poor knowledge score, and 107 participants (39.9%) had a good knowledge score. The total mean of attitude score was 16.23 ± 23.21 out of 26. A total of 148 participants (55.2%) had a poor attitude score and 120 participants (44.8%) had positive attitude. Participants with over 15 years of experience exhibited significantly higher odds of good knowledge (OR = 6.76, 95% CI = 1.03−44.21, *p* = 0.04). Participants who lived in the Western region had significantly lower odds of good knowledge (OR = 0.32, 95% CI = 0.14–0.71, *p* = 0.005). Nurses and respiratory therapists had significantly lower odds of having positive attitude (OR = 0.19, 95% CI = 0.09−0.42, *p* = 0.0001 and OR = 0.34, 95% CI = 0.16–0.75, *p* = 0.007, respectively). Participants who lived in the Western region had significantly lower odds of having positive attitude (OR = 0.42, 95% CI = 0.19–0.95, *p* = 0.005). **Conclusions:** Healthcare providers in Saudi Arabia demonstrated a moderate level of knowledge of ACP for COPD patients. This was accompanied by a moderately positive attitude towards this practice. Future studies should examine effective educational and professional interventions to enhance ACP practices.

## 1. Introduction

Globally, chronic obstructive pulmonary disease (COPD) is a considerable health problem; 480 million individuals aged 25 years and over were documented to be living with COPD in 2020, representing an estimated prevalence of 10.6% [1]. In addition, it is the fourth most common cause of mortality [2]. In 2021, about 5% of global mortality was caused by COPD [2]. COPD exacerbates unpredictably and causes hospitalization [3,4,5], which may lead to respiratory failure and require urgent decision-making [3,4,6,7,8]. To overcome these difficulties and improve outcomes in patients with COPD, advanced care planning (ACP) is critical [3,6,7,9,10,11].

Advanced care planning is a crucial legal and ethical aspect of palliative care [12,13,14], characterized as a constant, individual-focused communication approach that assists patients in understanding, consulting, and sharing their preferences, values, and objectives regarding prospective medical care [15,16]. ACP involves end-of-life (EOL) medical decisions, a healthcare agent, and advance directives [17]. Patients’ preferences, values, and objectives are usually documented in advance directives, which are legal documents that guide medical decision-making and prevent undesirable medical care in cases of incapability [18,19,20,21].

Advanced care planning for COPD patients could be associated with many benefits, including facilitating discussion of disease prognosis and diagnosis, reducing hospitalizations, and easing patients’ apprehensions about obtaining undesirable health interventions [22]. Earlier studies show that ACP for patients with COPD can improve symptoms and enhance patient–physician communication [23,24]. Therefore, policies recommend ACP for those patients [25,26]. In addition, there is evidence that ACP increases the usage of advance directives [26], which may improve patient outcomes. Previous studies indicate that advance directives reduce the cost and burden of healthcare and enhance the quality of EOL care without raising mortality [27,28]. However, ACP for COPD patients remains uncommon [29,30], as many patients and healthcare professionals avoid the topic and/or do not know when to start it [31,32]. In addition, there is evidence that the acceptance of advance directives is influenced by cultural differences between countries or regions within the same country [33]. Despite these, the general use of advance directives is increasing [34]. Healthcare providers should regularly discuss advance directives with their patients [35]. Thus, barriers to using advance directives for COPD patients must be addressed so that all patients receive equitable benefits.

Advance directives documenting is a valuable tool that regulates the physician–patient–family relationship, and is considered fundamental for understanding the patient’s perspective [36]. A recent literature review showed that appropriate healthcare professionals’ knowledge and attitudes concerning advanced directive care are important as suboptimal knowledge and attitudes could lead to the danger of improper utilization of voluntary assisted dying and the patient’s advance agreement [36]. Moreover, the previous literature showed that there is limited training related to this area [36]. Improving the attitudes, awareness, and knowledge of healthcare providers toward advance directives and/or ACP can increase the use of advance directives [37,38,39,40]. COPD represents a significant burden in Saudi Arabia, with prevalence increasing steadily since the 1990s [41]. The use of advance directives may help improve outcomes for those patients. This study aims to investigate healthcare providers’ knowledge, attitudes, and practices concerning advance directives for COPD patients in Riyadh, Saudi Arabia. Thus, it may aid in identifying strategies to enhance the use of advance directives for this population. To the best of our knowledge, there is no previous research that has examined healthcare providers’ knowledge, attitudes, and practices concerning advance directives for COPD patients in the Middle East. This gap in the knowledge offers the opportunity to examine the difference in advance directives for COPD patients due to the unique cultural and healthcare system in the Middle East.

## 2. Methods

### 2.1. Study Design

A cross-sectional study was carried out to assess the knowledge, attitudes, and practices of healthcare providers towards ACP for COPD patients in Riyadh between June and December 2024.

### 2.2. Study Population

The study population included physicians and nurses working in the intensive care unit (ICU) in hospitals across Riyadh, Saudi Arabia. Once the inclusion criteria were met, no further exclusion criteria were applied.

### 2.3. Participants Recruitment

Convenience sampling was employed to recruit the study participants for this research. Online survey and paper-based questionnaires were utilized in this research. For online survey, the questionnaire link was distributed across social media platforms including X, Facebook, and Snapchat. For the paper-based questionnaire, the study instrument was distributed physically to the study participants who met the inclusion criteria in the following hospitals in Riyadh, Saudi Arabia: King Fahad Medical City, King Faisal Specialist Hospital, Prince Mohammed Bin Abdulaziz Hospital, and Dr. Sulaiman Al Habib Hospital. The study aim and inclusion criteria were highlighted in the invitation letter of the questionnaire. Only those who met the inclusion criteria were requested to participate in the study.

### 2.4. Questionnaire Instrument

Data were collected via a self-administered survey consisting of a validated questionnaire. The questionnaire was adapted from the one used by AlFayyad et al. (2019) in their study on physicians’ and nurses’ knowledge and attitudes towards advance directives for cancer patients in Saudi Arabia [42]. The original questionnaire demonstrated acceptable levels of reliability and validity in the previous study population. The instrument included four sections: demographic data, healthcare professionals’ knowledge regarding advance directives, their attitudes toward advance directives, and their practices toward advance directives. The knowledge score was estimated by giving a score of 1 for each correct answer, with a maximum score of 12. Similarly, the attitude score was estimated giving a score of 1 for each correct answer, with a maximum score of 26. The higher the score, the higher the level of knowledge and the more positive the attitude towards advance directives for COPD patients.

### 2.5. Piloting Phase

At first, the face validity of the questionnaire tool was checked by two clinicians who were experts in the area of COPD. The two clinicians confirmed that the questionnaire tool items measure healthcare providers’ knowledge, attitude, and practice concerning advance directives for COPD patients. The questionnaire tool used in this research was piloted on a small number of participants (15 participants) who met the inclusion criteria. The study participants were asked about the clarity of the questionnaire tool, and they confirmed that questionnaire items were clear and easy to understand. The questionnaire used in this research demonstrated good internal consistency with Cronbach’s alpha measure of 0.812.

### 2.6. Ethical Approval

Ethical approval for this research was obtained from the Institutional Review Board (IRB) in Al-Imam Muhammad Ibn Saud Islamic University (project number 655/2024) (on 20 June 2024).

### 2.7. Statistical Analysis

Descriptive statistics such as the frequency and percentage were used to illustrate categorical variables; the mean and the standard deviation (SD) were used to illustrate the continuous variables due to the normality of the data, which was checked using histogram, skewness, and kurtosis measures. The Analysis of Variance (ANOVA) test and the independent *t*-test were performed when applicable to examine the difference in continuous variables. Tukey’s post hoc test was applied for multiple-group comparisons. Multiple logistic regression was performed to assess the factors associated with good knowledge and positive attitude. The results from the regression analyses were presented as odds ratios (ORs) with 95% confidence intervals (CIs). The level of significance was defined as α = 0.05. All calculations and analyses were carried out with the Statistical Package of Social Sciences, Version 29.0.

## 3. Results

A total of 268 participants were included in the analysis. Regarding age distribution, the majority were aged between 25 and 34 years (n = 112; 41.8%), followed by those aged between 35 and 44 years (n = 92, 34.3%). Most participants were male (n = 204; 76.1%), while females constituted 64 participants (23.9%). The majority of the sample were Saudi nationals (n = 243; 90.7%). Regarding occupation, 128 participants (47.8%) were nurses, and 65 (24.3%) were physicians. Additional details about demographic characteristics are provided in Table 1.

The majority of participants (n = 234; 87.3%) were aware that an advance directive is a legal document that informs the physician earlier about patients’ wishes about future healthcare if they become mentally incompetent. Regarding the types of advance directives, a total of 197 participants (73.5%) recognized the living will and the durable power of attorney for healthcare. Over half of the participants (n = 154; 57.5%) understood that living will governs specific future healthcare decisions when a patient becomes incapable of making decisions. Additionally, 147 participants 1954.9%) knew that a durable power of attorney allows a designated person to make healthcare decisions for a patient who is incapable. Additional details about knowledge are provided in Table 2.

The majority of participants (n = 208; 77.6%) agreed that the advanced directive should be discussed with every patient regardless of their diagnosis. Additionally, 170 participants (63.4%) believed that discussing the advanced directive is particularly important for patients diagnosed with life-threatening diseases. Regarding its impact, 126 participants (47.0%) agreed that the advanced directive could help reduce a decisional catastrophe regarding end-of-life care. Additionally, 154 participants (57.5%) expressed confidence in making treatment choices during catastrophic situations when guided by an advanced directive. Additional details about attitude are provided in Table 3.

The majority of participants (n = 201; 75.0%) reported that their healthcare facility provides standardized advance directive forms for patients. Most respondents (n = 182, 67.9%) felt supported by their colleagues in discussing advanced care planning with patients and families. Regarding routine initiation of advanced care planning discussions with patients with COPD, a total of 117 participants (43.7%) reported performing this practice. Additional details about participants practice are provided in Table 4.

The total mean of knowledge score was 6.96 ± 2.22 out of 12. As the table shows, Saudi nations reported a significant lower knowledge score mean (6.82 ± 2.16) compared to non-Saudi nations (8.28 ± 2.49) (*p* = 0.002). Physicians reported a significant higher knowledge score mean (7.51 ± 2.65) compared to nurses (6.60 ± 2.08) (*p* = 0.002). Furthermore, participants with experience over 15 years reported a significant higher knowledge score (9.13 ± 3.07) compared to participants with less than 5 years of experience (6.82 ± 2.80) (*p* = 0.001). The total mean of attitude score was 16.23 ± 23.21 out of 26. As the table shows, Saudi nations reported a significant lower attitude score mean (15.96 ± 4.82) compared to non-Saudi nations (18.84 ± 3.97) (*p* = 0.004). Physicians reported a significant higher attitude score mean (18.34 ± 5.56) compared to nurses (15.27 ± 4.26) (*p* = 0.001). Furthermore, participants with experience over 15 years reported a significant higher knowledge score (19.67 ± 4.40) compared to participants with less than 5 years of experience (16.80 ± 6.30) (*p* = 0.001). Additional details about knowledge and attitude scores stratified by the demographics are provided in Table 5.

A total of 161 participants (60.1%) had a poor knowledge score and 107 participants (39.9%) had a good knowledge score. A total of 148 participants (55.2%) had a poor attitude score and 120 participants (44.8%) had positive attitude. A multiple logistic regression model was used to assess the factors affecting the knowledge and attitude levels. Participants with over 15 years of experience had significantly higher odds of good knowledge (OR = 6.76, 95% CI = 1.03−44.21, *p* = 0.04). Participants who lived in the Western region had significantly lower odds of good knowledge (OR = 0.32, 95% CI = 0.14–0.71, *p* = 0.005). Regarding the attitude level, nurses and respiratory therapists had significantly lower odds of having positive attitude (OR = 0.19, 95% CI = 0.09−0.42, *p* = 0.0001, and OR = 0.34, 95% CI = 0.16–0.75, *p* = 0.007, respectively). Participants who lived in the Western region had significantly lower odds of having positive attitude (OR = 0.42, 95% CI = 0.19–0.95, *p* = 0.005). Further details about factors affecting the knowledge and attitude score are provided in Table 6.

## 4. Discussion

Comprehending the awareness, attitudes, and knowledge of advance directives aids in determining factors that need to be modified in this area, which contributes to developing more effective approaches [43]. The current investigation found that the majority of participants (n = 234; 87.3%) were aware that an advance directive is a legal document that informs the physician earlier about patients’ wishes about future healthcare if they become mentally incompetent. Consistent with our findings, previous studies indicate a high awareness of advance directives among healthcare providers or patients. In Ohio, the accurate answer about the description of advance directives among nurses was between 95% and 99% [44]. Research conducted among Swiss adults aged 55 and over and Portuguese adults showed that approximately 79% and 77% had heard of advance directives, respectively [45,46]. Approximately 83% of nurses and physicians in an adult intensive care unit in Spain believe that healthcare professionals use advance directives in decision-making [47]. In Saudi Arabia, most cancer nurses (82%) and cancer physicians (65%) described advance directives correctly [42].

On the contrary, awareness of advance directives was low among participants in other previous studies. In Brazil, 62% of health professionals [48], 55% of psychiatrists [49], and 77% of medical students, elderly caregivers, and professors of medicine [50] did not know what advanced directives are. In China, awareness of advance directives among healthcare providers and cancer patients was low [51]. The difference in awareness level may be due to a disparity in cultural factors, educational programs and activities, legal aspects, and demographic characteristics. These aspects need to be assessed to improve awareness of advance directives.

In this study, most participants (n = 208; 77.6%) agreed that advanced directives should be discussed with every patient regardless of their diagnosis. This finding aligns with recommendations supporting proactive, routine discussions about advanced directives. Advance care directives are best discussed with all adults when they are well before they develop any acute condition, as they will be less stressed to discuss them and because unexpected EOL situations can occur at any time in their lives [52]. Many clinics, such as the Cleveland Clinic, recommend that every adult (age 18 and older) have an advance directive in their electronic health record [53]. Several prior studies have suggested that ACP should be proactive, incorporated into regular care, and updated when patients’ health status changes [54,55]. For COPD, earlier research has recommended proactively implementing ACP when the disease is stable to maximize effectiveness for COPD patients [56].

Despite the high awareness of advance directives among healthcare providers in our study, this does not necessarily mean that knowledge and attitude will be good. The results of this study show that most participants had poor knowledge and attitude; a total of 161 participants (60.1%) had a poor knowledge score, and a total of 148 participants (55.2%) had a poor attitude score. Confirming this, most healthcare providers and cancer patients in China had low awareness of advance directives, but after explaining the information about them, they showed positive attitudes toward them [51].

In line with our findings, a previous study in Spain found that knowledge and attitudes towards advance directives among health professionals in emergency medical services were low, with most of them considering their understanding of these directives inadequate, and most did not integrate them into their regular practice [57]. Similarly, knowledge and attitudes toward advance directives among emergency healthcare providers in South Korea were moderately low [58]. According to a prior study, healthcare professionals in Würzburg need further training on advance directives due to their lack of knowledge of the practical and ethical aspects of advance directives [59]. In Brazil, knowledge of advance directives was less than ideal in a teaching hospital [48] and was low among medical students, elderly caregivers, and professors of medicine [50]. These findings demonstrate that low knowledge and attitude regarding advance directives among healthcare providers and patients are a significant problem in many countries. Therefore, conducting more educational and training activities regarding advance directives for healthcare providers and individuals is a critical requirement due to the importance of these directives in maintaining patient independence [59].

It is worth noting here that the level of knowledge about advance directives does not necessarily correlate with attitudes toward advance directives. Two previous investigations in Spain indicated that nurses, physicians, and professionals had positive attitudes toward advance directives, although their knowledge of them was low [39,60]. An earlier study found that electronic reminders and education for physicians regarding the implementation of advance directives for patients with chronic diseases, including COPD, had a positive but limited effect on their implementation [61]. Hence, in addition to good training for healthcare providers about advance directives, all institutions and organizations should become cognizant of the importance of advance directives in meeting patients’ expectations and needs, reducing service disruptions, and improving healthcare quality [57]. Further research is needed to address other gaps in advance directive implementation, such as healthcare providers’ time constraints.

In our study, participants with over 15 years of experience had significantly higher odds of knowledge (OR = 6.76, 95% CI = 1.03−44.21, *p* = 0.04). Participants who lived in the Western region had significantly lower odds of knowledge (OR = 0.32, 95% CI = 0.14–0.71, *p* = 0.005) and significantly lower odds of attitude (OR = 0.42, 95% CI = 0.19–0.95, *p* = 0.005). Nurses and respiratory therapists had significantly lower odds of attitude (OR = 0.19, 95% CI = 0.09−0.42, *p* = 0.0001 and OR = 0.34, 95% CI = 0.16–0.75, *p* = 0.007, respectively). Consistent with our findings, in Spain, healthcare professionals’ experience with advance directives was associated with their level of knowledge [39]. Previous research demonstrated that practical experience and engagement with advance directives are associated with higher level of knowledge [39]. Experience and knowledge are connected to each other and form a cycle of continuous improvement.

In a previous study among healthcare professionals in Würzburg, knowledge of advance directives was higher among males with more experience [59]. Similarly, in South Korea, demographic and experience differences influenced knowledge, with older and more experienced emergency healthcare providers having higher scores in knowledge of advance directives [58]. Experience appears to impact attitudes, and demographic factors such as gender and age also affect the degree of knowledge in previous studies. Thus, tailored interventions that regard healthcare providers’ experience and demographic differences could enhance their knowledge of advance directives.

The low knowledge and attitudes about advance directives among our participants from the Western region may be due to cultural differences or differences in laws relating to advance directives in that region. Evidence shows that laws on advance directives vary from region to region [35] and that cultural differences influence the acceptance of advance directives across areas [33]. Consistent with this, in a previous study of Australian and Irish doctors, knowledge was higher among Australian doctors [62]. In another study, awareness and completion of advance directives varied significantly across different parts of Switzerland, depending on the language spoken in the region [45]. Accordingly, cultural differences and laws regarding advance directives must be considered when planning any strategy to enhance their implementation.

Regarding our finding that nurses and respiratory therapists had significantly lower odds of attitude about advance directives, this finding is also consistent with several previous studies that have indicated that healthcare providers’ attitudes vary across their areas of expertise and profession [42,63,64]. Nurses and respiratory therapists’ lower odds towards advance directives could be justified by the fact that they frequently encounter life-saving situations due to their work in ICU settings where their focus is on life-saving rather than end-of-life planning. Other possible justifications include personal beliefs, cultural barriers, and ethical considerations. Both culture and laws influence individuals’ attitude towards advance directives. A previous study by Solkowski et al. reported that higher belief in God’s role at the end of life (EOL) was associated with a higher preference for life-prolonging measures. At the same time, those with a higher belief in God’s role at EOL had lower odds of AD completion [65]. A literature review published in 2019 showed that physicians are influenced by legal concerns while implementing the directives and that legal concerns are extremely important for their decision concerning advance directives [66].

Although the primary caregivers of patients with COPD are respiratory physicians, their opinions on ACP have not been widely regarded [11,67]. Ultimately, all healthcare providers, whether treating patients with chronic diseases such as COPD or others, should be adequately trained and provided with adequate and up-to-date information about advance directives and how to discuss them with their patients to achieve a better quality of care and quality of life for all patients and individuals. Moreover, practical recommendations include hospitals having ACP discussion in their admission and discharge routine of care. In addition, hospitals and decision-makers in the healthcare sector should regulate ACP through the establishment of ethics committees that regulate ACP-related conflicts in coordination with legal and religious authorities.

This research has limitations. The cross-sectional study design limited the ability to examine causality among the study variables. The convenience sampling technique utilized in this research could have increased the possibility of selection bias as it might not have covered the whole targeted study population who do not have access to the social media platforms utilized in this research, which might have affected the generalizability of our study findings. However, in order to decrease the possibility of selection bias, this research utilized both online survey and paper-based questionnaires.

## 5. Conclusions

Healthcare providers in Saudi Arabia demonstrated a moderate level of knowledge of ACP for COPD patients. This was accompanied by a moderately positive attitude towards this practice. Future studies should aim to examine effective educational and professional interventions to enhance ACP practices.

## Figures and Tables

**Table 1 healthcare-13-00771-t001:** Demographic characteristics of the participants.

Variable		Frequency	Percentage
Age (years)	<25	17	6.3%
	25–34	112	41.8%
	35–44	92	34.3%
	45–54	42	15.7%
	55 and older	5	1.9%
Sex	Male	204	76.1%
	Female	64	23.9%
Nationality	Non-Saudi	25	9.3%
	Saudi	243	90.7%
Healthcare role	Physician	65	24.3%
	Nurse	128	47.8%
	Respiratory therapist	75	28.0%
Experience (years)	0–5	56	20.9%
	6–10	112	41.8%
	11–15	85	31.7%
	15 and above	15	5.6%
Region	Central	62	23.1%
	Western	94	35.1%
	North	22	8.2%
	South	26	9.7%
	East	64	23.9%
Place of practice	University hospital	74	27.6%
	MOH	51	19.0%
	Private	19	7.1%
	Medical city	41	15.3%
	Military	83	31.0%
Specialty	Non-physician	204	76.1%
	Family medicine	16	6.0%
	Intensivist	21	7.8%
	Internal medicine	8	3.0%
	Pulmonologist	19	7.1%
Please indicate your level of experience	Non-physician	204	76.1%
	Consultant	14	5.2%
	Fellow	15	5.6%
	Registrar	7	2.6%
	Resident	11	4.1%
	Senior registrar	17	6.3%

**Table 2 healthcare-13-00771-t002:** The knowledge about advance directive for COPD patients.

	No		Yes	
	Frequency	Percentage	Frequency	Percentage
“An advance directive is a legal document that informs the physician about patients’ wishes earlier about future healthcare if they become mentally incompetent.”	34	12.7%	234	87.3%
“The types of advance directives are the living will and the durable power of attorney for healthcare.”	71	26.5%	197	73.5%
“The living will is a document that aims to govern specific future healthcare decisions merely when a patient becomes incapable to make decisions on their own.”	114	42.5%	154	57.5%
“A durable power of attorney for health is an official document in which patient designates a person to be their proxy to make all their healthcare decisions if they become incapable.”	121	45.1%	147	54.9%
“An advance directive will influence the type or quality of patient care while they can express their decisions. It only becomes effective when they mentally can do so.”	101	37.7%	167	62.3%
“In the advance directives, the patient can decide whether or not to use life-sustaining machines, like a mechanical ventilator and dialysis.”	132	49.3%	136	50.7%
“In the advance directives, the patient can decide whether or not to have a CPR or DNR.”	144	53.7%	124	46.3%
“In the advance directives, the patient can decide whether or not to withhold nutrition and hydration.”	158	59.0%	110	41.0%
“In the advance directives, the patient can decide the place of terminal care and death.”	137	51.1%	131	48.9%
“The most appropriate time to discuss advanced directives is when the patient is terminally or seriously ill.”	148	55.2%	120	44.8%
“In an effective advanced directive communication, it is imperative to ask the patient to nominate a principal person as a healthcare proxy.”	102	38.1%	166	61.9%
“It is imperative to include the patient’s healthcare proxy in the discussion of the advance directive.”	89	33.2%	179	66.8%

**Table 3 healthcare-13-00771-t003:** The attitude about advance directive for COPD patients.

	No		Yes	
	Frequency	Percentage	Frequency	Percentage
“The advanced directive has to be discussed with every patient irrespective of their diagnosis”	60	22.4%	208	77.6%
“Discussion of the advanced directive is imperative to patients who are diagnosed with life-threatening diseases”	98	36.6%	170	63.4%
“The advanced directive could lessen the end-of-life care decisional catastrophe”	142	53.0%	126	47.0%
“In a catastrophic situation, you would have more confidence in the treatment choices if directed by an advance directive”	114	42.5%	154	57.5%
“You would worry less about legal consequences of limiting treatment if you were following an advance directive”	120	44.8%	148	55.2%
“Discussion of advanced directive could end patients’ sense of hope”	46	17.2%	222	82.8%
“Discussion of advanced directive could improve patients’ and families’ satisfaction with end-of-life care”	106	39.6%	162	60.4%
“Advanced directive reduces the use of futile/unnecessary care at the end of life”	116	43.3%	152	56.7%
“Discussion of the advanced directive is the physician’s responsibility”	130	48.5%	138	51.5%
“Practicing advanced directive could be consistent with patient-centered care standards in your healthcare institution”	106	39.6%	162	60.4%
“Most of your patients are willing to know their diagnosis, prognosis, and care options”	95	35.4%	173	64.6%
“Most patients with end-stage diseases are willing to communicate their wishes for end-of-life care”	117	43.7%	151	56.3%
“In your culture, it feels easy when discussing matters related to the end of life with patients and their families”	22	8.2%	246	91.8%
“In your culture, discussion of an advance directive would produce a more confrontational relationship with the patient”	129	48.1%	139	51.9%
“A prospective problem with advance directives is that patients’ families could change their minds about treatment when their patient becomes terminally ill”	103	38.4%	165	61.6%
“In your culture, it feels easy when discussing advanced directives with patients with progressive diseases”	25	9.3%	243	90.7%
“The advanced directive in the long term reduces the cost of unnecessary treatment/care”	111	41.4%	157	58.6%
“Advance directive documents could be useful in your institution”	125	46.6%	143	53.4%
“Your administration/colleagues would support the practice of advanced directives”	113	42.2%	155	57.8%
“The advance directive may be a relief for families in some circumstances”	121	45.1%	147	54.9%
“The advance directive might be culturally accepted and established”	115	42.9%	153	57.1%
“The advance directive does not interfere with Islamic regulations”	122	45.5%	146	54.5%
“The advance directive can be applied in your institution if legalized”	111	41.4%	157	58.6%
“The advance directive in the long term positively affects the cost of total care and saves medical expenditures”	86	32.1%	182	67.9%
“The advance directive can improve and facilitate the discharge plan process”	109	40.7%	159	59.3%
“You would recommend your healthcare institution to adopt the practice of advance directives”	77	28.7%	191	71.3%

**Table 4 healthcare-13-00771-t004:** Practices and perception regarding advance care planning for COPD patients.

	I Do Not Know		No		Yes	
	Frequency	Percentage	Frequency	Percentage	N	%
“Does your healthcare facility provide standardized advance directive forms specifically for patients”	18	6.7%	49	18.3%	201	75.0%
“My colleagues support me in discussing advanced care planning with patients and families”	18	6.7%	68	25.4%	182	67.9%
“In my practice, I routinely initiate advanced care planning discussions with patients with COPD”	26	9.7%	125	46.6%	117	43.7%
“In my practice, I routinely follow up advanced care planning discussions, when appropriate, with patients with COPD”	20	7.5%	114	42.5%	134	50.0%
“In my practice, I have had advanced care planning discussions with more than 50% of patients with COPD”	37	13.8%	113	42.2%	118	44.0%
“In my practice, I routinely talk with patients and families about palliative and hospice care options when appropriate to patients’ disease status”	18	6.7%	79	29.5%	171	63.8%
“I feel comfortable discussing issues related to death and dying with patients and their families”	22	8.2%	101	37.7%	145	54.1%
“I feel comfortable discussing advanced care planning with patients with COPD”	43	16.0%	112	41.8%	113	42.2%
“I have sufficient knowledge about how to conduct advanced care planning conversations with patients and their families”	21	7.8%	79	29.5%	168	62.7%
“I feel confident in my ability to communicate“bad news””	31	11.6%	101	37.7%	136	50.7%
“Have you ever been involved in a situation where implementing an advance directive for a COPD patient presented significant challenges”	19	7.1%	76	28.4%	173	64.6%

**Table 5 healthcare-13-00771-t005:** The knowledge score stratified by the demographic characteristics.

		Knowledge			Attitude		
		Mean	SD	*p* Value	Mean	SD	*p* Value
Age (years)	<25	6.82	2.98	0.60	17.18	5.50	0.51
	25–34	7.01	2.26		16.60	5.30	
	35–44	6.73	1.92		15.64	4.02	
	45–54	7.29	2.36		16.33	4.71	
	55 and older	7.80	3.11		14.60	5.94	
Gender	Male	7.00	2.06	0.54	16.19	4.50	0.80
	Female	6.81	2.71		16.36	5.76	
Nationality	Non-Saudi	8.28	2.49	0.002 *	18.84	3.97	0.004 *
	Saudi	6.82	2.16		15.96	4.82	
Healthcare role	Physician	7.51	2.65	0.02 *	18.34	5.56	0.001 *
	Nurse	6.60	2.08		15.27	4.26	
	Respiratory therapist	7.09	1.97		16.03	4.51	
Experience (years)	0–5	6.82	2.80	0.001 *	16.80	6.30	0.01 *
	6–10	6.94	1.85		15.92	4.19	
	11–15	6.69	1.92		15.65	4.31	
	15 and above	9.13	3.07		19.67	4.40	
Region	Central	7.87	2.43	0.001 *	19.10	4.95	0.001 *
	Western	6.56	1.84		15.71	3.89	
	North	6.77	2.37		13.27	4.60	
	South	7.81	2.70		17.23	6.42	
	East	6.38	1.97		14.81	3.85	
Place of practice	University hospital	6.73	2.06	0.01 *	15.50	4.64	0.01 *
	MOH	7.43	2.50		17.69	5.27	
	Private	8.16	2.39		18.32	4.63	
	Medical city	7.20	2.36		16.44	5.66	
	Military	6.48	1.96		15.40	3.95	

* The mean difference is significant at the 0.05 level.

**Table 6 healthcare-13-00771-t006:** Logistic regression analysis of demographic characteristics and knowledge and attitude level.

Variable		Knowledge		Attitude	
		OR (95% CI)	*p* Value	OR (95% CI)	*p* Value
Age (years)	<25	Reference		Reference	
	25–34	0.95 (0.26–3.44)	0.940	0.61 (0.17–2.20)	0.453
	35–44	1.06 (0.25–4.37)	0.940	0.30 (0.07–1.24)	0.096
	45–54	0.83 (0.16–4.15)	0.816	0.26 (0.05–1.31)	0.102
	55 and older	0.20 (0.01–4.69)	0.319	0.09 (0.00–1.83)	0.118
Gender	Male	Reference		Reference	
	Female	0.97 (0.50–1.86)	0.923	0.83 (0.43–1.62)	0.592
Nationality	Non-Saudi	Reference		Reference	
	Saudi	0.66 (0.23–1.86)	0.429	0.67 (0.23–1.93)	0.454
Healthcare role	Physician	Reference		Reference	
	Nurse	0.62 (0.29–1.31)	0.214	0.19 (0.09–0.42)	0.000
	Respiratory therapist	1.00 (0.46–2.20)	0.992	0.34 (0.16–0.75)	0.007
Experience (years)	0–5	Reference		Reference	
	6–10	0.99 (0.42–2.30)	0.980	1.28 (0.55–3.00)	0.569
	11–15	0.65 (0.24–1.75)	0.397	2.07 (0.78–5.49)	0.145
	15 and above	6.76 (1.03–44.21)	0.046	3.54 (0.60–20.74)	0.161
Region	Central	Reference		Reference	
	Western	0.32 (0.14–0.71)	0.005	0.42 (0.19–0.95)	0.038
	North	0.50 (0.15–1.61)	0.245	0.22 (0.06–0.76)	0.017
	South	1.42 (0.48–4.14)	0.526	0.23 (0.07–0.72)	0.012
	East	0.42 (0.17–1.03)	0.058	0.24 (0.09–0.62)	0.003
Place of practice	University hospital	Reference		Reference	
	MOH	1.20 (0.51–2.82)	0.678	1.56 (0.66–3.67)	0.307
	Private	2.61 (0.76–8.99)	0.127	1.24 (0.36–4.31)	0.732
	Medical city	1.28 (0.54–3.06)	0.576	0.96 (0.40–2.32)	0.933
	Military	0.85 (0.41–1.78)	0.674	1.02 (0.49–2.12)	0.948

## Data Availability

The datasets used and/or analyzed during the current study are available from the corresponding author on reasonable request.

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
