# Peer review of "Knowledge, Attitudes, and Practices of Healthcare Providers Towards Advance Directive for COPD Patients in Riyadh, Saudi Arabia"

_healthcare, 2025, doi:10.3390/healthcare13070771_

Round 1
Reviewer 1 Report
Comments and Suggestions for Authors
The manuscript (healthcare-3512985) explores around awareness of healthcare providers and utilization of innovative approaches, specifically in case of COPD patients. Although, the study is interesting but there are number of major areas that needs to improved.The authors may look into following specific comments that are important to be:
- The introduction section should give details regarding research gap with regards to advance directives, particularly how this study is different from previous research.
- The following statement should be supported with relevant reference; "COPD exacerbates unpredictably, which may lead to respiratory failure and require urgent decision-making".
- The impact of selection bias should discussed with respect to sampling technique.
- Did authors conducted pilot study before conducting full implementation and more details regarding validity and reliability should be provided.
- How did missing data were handled during analysis as this is an important factor for self-adminsitered suveys.
- Authors should also add contexting details to mean knowledge and attitude scores, also how they compare with similar studies in other regions.
- The linkage between experience and knowledge should be discussed more comprehensively.
- Discussion section should also include the details on whether nurses and respiratory therapists had lesser chances of exhibiting positive attitude towards advance directives. Also include probable reasons for it.
- The cutlurat and legal side of advance directives needs to be discuss in more details.
- Authors should expand on practical recommendations, such as inclusion of discussion about ACP for COPD patients in hospital policies.
- Author should also include whether intervention study should be included to test education programs for healthcare providers.
Author Response
Dear Reviewer 1,
Author's reply to Reviewer 1's comments:
The manuscript (healthcare-3512985) explores around awareness of healthcare providers and utilization of innovative approaches, specifically in case of COPD patients. Although, the study is interesting but there are number of major areas that needs to improved. The authors may look into following specific comments that are important to be:
Comment: 1. The introduction section should give details regarding research gap with regards to advance directives, particularly how this study is different from previous research.
- Thank you for this comment, we have now addressed this point in the introduction, see lines 77-83 and lines 90-95.
Comment: 2. The following statement should be supported with relevant reference; "COPD exacerbates unpredictably, which may lead to respiratory failure and require urgent decision-making".
- Thank you for this comment, we have now clarified this sentence further and added relevant supporting references, see line 49.
Comment: 3. The impact of selection bias should discussed with respect to sampling technique.
- Thank you for this comment, we have now added new paragraph highlighting the study limitations including the possibility of selection bias, see lines 349-355.
Comment: 4. Did authors conducted pilot study before conducting full implementation and more details regarding validity and reliability should be provided.
- Thank you for this comment, we have now added new paragraph highlighting the piloting phase for this study, see lines 129-137.
Comment: 5. How did missing data were handled during analysis as this is an important factor for self-adminsitered suveys.
- Thank you for this comment. In our research we did not encounter missing data as study participants completed all questionnaire items.
Comment: 6. Authors should also add contexting details to mean knowledge and attitude scores, also how they compare with similar studies in other regions.
- Thank you for this comment. We have now added further details on the procedure of estimating the knowledge and attitude scores, see lines 124-128. The procedure followed for estimating the mean knowledge and attitude score was similar to the one followed by the original research by AlFayyad et al., 2019.
Comment: 7. The linkage between experience and knowledge should be discussed more comprehensively.
- Thank you for this comment, we have now discussed this point in the discussion section, see lines 302-305.
Comment: 8. Discussion section should also include the details on whether nurses and respiratory therapists had lesser chances of exhibiting positive attitude towards advance directives. Also include probable reasons for it.
- Thank you for this comment, we have now discussed this point in the discussion section, see lines 327-331.
Comment: 9. The cutlurat and legal side of advance directives needs to be discuss in more details.
- Thank you for this comment, we have now discussed this point in the discussion section, see lines 331-338.
Comment: 10. Authors should expand on practical recommendations, such as inclusion of discussion about ACP for COPD patients in hospital policies.
- Thank you for this comment, we have now discussed this point in the discussion section, see lines 339-348.
Comment: 11. Author should also include whether intervention study should be included to test education programs for healthcare providers.
- Thank you for this comment, we already added this recommendation for future research, see lines 359-360.
Reviewer 2 Report
Comments and Suggestions for Authors
I read with interest the paper titled "Knowledge, Attitude and Practice of Healthcare Providers towards Advance Directive for COPD patients in Riyadh, Saudi Arabia"
I have few comments that could enhance the manuscript. The article is well contructed and easy to read.
1. Convenience sample was used. However, it will be good to further explain how do you reach 268 participants? Were that all that enrolled the study, or did you stop when you reach a specific quantity of participants?
2. Was the sample size calculated? Despite of being a convenience sample, did authors feel that this is a representative of the healthcare professionals surveyed?
3. Why authors use two types of questionnaire, namely online and paper based? This represents a bias in the collection of information, that should be further explored.
4. What were the adapted changes that were performed in the previous questionnaire that was used as basis to that one?
5. Authors state that "The original questionnaire demonstrated acceptable levels of reliability and validity in the previous study population" (line 107) - did you test for the updated one? If yes, please share the results.
6. Did the authors made assumption tests for the requirements of parametric tests? If yes, they should be stated. If not, they should be performed to ensure data aproximation to normal distribution
7. Gender should be "sex" instead. Gender is a social construct, while sex is what was surveyed.
8. A section with limitations (or at least a paragraph in the end of the discussion) should be added.
Author Response
Dear Reviewer 2,
The author's reply to Reviewer 2's comments:
Reviewer 2: I read with interest the paper titled "Knowledge, Attitude and Practice of Healthcare Providers towards Advance Directive for COPD patients in Riyadh, Saudi Arabia"
I have few comments that could enhance the manuscript. The article is well contructed and easy to read.
- Comment: Convenience sample was used. However, it will be good to further explain how do you reach 268 participants? Were that all that enrolled the study, or did you stop when you reach a specific quantity of participants?
- Thank you for this comment. We collected data for six months as per our plan. We aimed to reach a sample size similar to the one achieved by the original study by AlFayyad et al., 2019. We included all participants who meet the inclusion criteria and agreed to participate in the study.
- Comment: Was the sample size calculated? Despite of being a convenience sample, did authors feel that this is a representative of the healthcare professionals surveyed?
- Thank you for this comment, as highlighted above, we collected data for six months as per our plan. We aimed to reach a sample size similar to the one achieved by the original study by AlFayyad et al., 2019. Moreover, due to the exploratory nature of the study, sample size calculation was deemed not necessary. We added the limitation of the use of convenience sampling technique to address the reviewer comment further, see lines 349-355.
- Comment: Why authors use two types of questionnaire, namely online and paper based? This represents a bias in the collection of information, that should be further explored.
- Thank you for this comment. In order to decrease the possibility of selection bias, this research utilized both online survey and paper-based questionnaire, we have now highlighted this point in the study limitations, see lines 349-355.
- Comment: What were the adapted changes that were performed in the previous questionnaire that was used as basis to that one?
- Thank you for this comment. The original research by AlFayyad et al., 2019 was developed to examine. We expanded the demographic characteristics being examined for the study participants to include healthcare professional role not only physicians, different years of experience range, region of practice, place of practice, and specialty.
- Comment: Authors state that "The original questionnaire demonstrated acceptable levels of reliability and validity in the previous study population" (line 107) - did you test for the updated one? If yes, please share the results.
- Thank you for this comment. We have now addressed the reviewer comment and added Cronbach’s Alpha measure for our study which was 0.812 and demonstrated good internal consistency, see lines 136-137.
- Comment: Did the authors made assumption tests for the requirements of parametric tests? If yes, they should be stated. If not, they should be performed to ensure data aproximation to normal distribution.
- Thank you for this comment. We have now addressed the reviewer comment and highlighted that we have checked the normality of the data before deciding to use parametric tests, see lines 144-146.
- Comment: Gender should be "sex" instead. Gender is a social construct, while sex is what was surveyed.
- Thank you for this comment. We have now addressed the reviewer comment.
- Comment: A section with limitations (or at least a paragraph in the end of the discussion) should be added.
- Thank you for this comment. We have now addressed the reviewer comment, see lines 349-355.
Round 2
Reviewer 1 Report
Comments and Suggestions for Authors
The manuscript can be accepted in its present form.